# On the Dimensionality of Sentence Embeddings

**Hongwei Wang, Hongming Zhang, Dong Yu**

Tencent AI Lab Seattle

{hongweiw, hongmzhang, dyu}@global.tencent.com

## Abstract

Learning sentence embeddings is a fundamental problem in natural language processing. While existing research primarily focuses on enhancing the quality of sentence embeddings, the exploration of sentence embedding dimensions is limited. Here we present a comprehensive and empirical analysis of the dimensionality of sentence embeddings. First, we demonstrate that the optimal dimension of sentence embeddings is usually smaller than the default value. Subsequently, to compress the dimension of sentence embeddings with minimum performance degradation, we identify two components contributing to the overall performance loss: the encoder's performance loss and the pooler's performance loss. Therefore, we propose a two-step training method for sentence representation learning models, wherein the encoder and the pooler are optimized separately to mitigate the overall performance loss in low-dimension scenarios. Experimental results on seven STS tasks and seven sentence classification tasks demonstrate that our method significantly improves the performance of low-dimensional sentence embeddings.

## 1 Introduction

Learning sentence representation is a fundamental problem in natural language processing. Sentence embeddings represent sentences as fixed-length vectors, which can be used in various downstream tasks, such as semantic textual similarity (STS) (Agirre et al., 2012, 2013; Marelli et al., 2014), information retrieval (Mitra et al., 2017; Karpukhin et al., 2020; Thakur et al., 2021), and sentiment analysis (Pang and Lee, 2005; Hu and Liu, 2004; Pang and Lee, 2004).

Existing work usually focuses on improving the quality of sentence embeddings by introducing novel model architectures or training strategies (Reimers and Gurevych, 2019; Liu et al., 2021; Gao et al., 2021; Chuang et al., 2022; Su et al.,

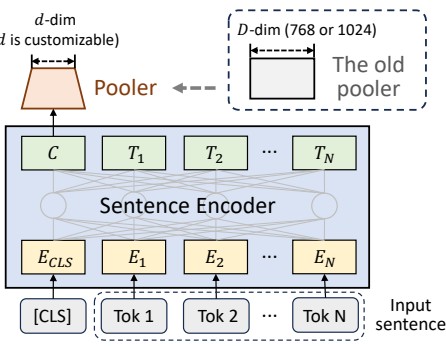

Figure 1: The proposed architecture of a sentence representation learning model. The dimension of the pooler's fully connected layer is changed from $D \times D$ to $D \times d$, where $D$ is the hidden state dimension (768 for base models and 1,024 for large models), and $d$ is the customizable sentence embedding dimension. The remaining part of the model (sentence encoder) is unchanged.

2022). However, the exploration of sentence embedding dimensions remains limited. These sentence representation learning models typically employ the default dimension of the model's hidden states as the dimension of sentence embeddings (e.g., 768 for BERT$_{base}$-like models and 1,024 for BERT$_{large}$-like models). Nonetheless, the dimension plays a critical role in sentence embeddings, and many research questions regarding its impact on sentence embeddings remain unanswered. For instance, does the default dimension yield the best performance? Can the dimension of sentence embeddings be reduced to mitigate time and memory burdens in practical applications? Furthermore, how can we maintain the performance of sentence embeddings when their dimension is reduced?

In this paper, we aim to answer the above questions through a comprehensive and empirical study of the dimensionality of sentence embeddings. Unlike conventional post-processing dimension reduction methods, we propose a direct modification of the output dimension of the pooler in sentence representation learning models, as illustrated in Figure

1. This approach enables us to generate sentence embeddings of any desired dimension while imposing minimal computational overhead. Subsequently, we evaluate sentence embeddings with various dimensions across various downstream tasks. The findings indicate that the optimal dimension for sentence embeddings tends to be smaller than the default value used in the literature.

Our findings also indicate a significant decline in the performance of sentence embeddings when the dimension is reduced beyond the optimal value. Therefore, we investigate whether the model's performance can be sustained in these low-dimension scenarios. This allows us to utilize sentence embeddings with even smaller dimensions in practical applications to reduce time and memory overhead further. Interestingly, we find that the model's performance deterioration in low-dimension scenarios is not solely attributed to the decrease of the pooler's output dimension, but also to the degradation in the quality of the sentence encoder's output. As a result, the performance loss can be divided into two components: the loss caused by the encoder and the loss caused by the pooler. We then propose a two-step training algorithm to mitigate the two aspects of the performance loss separately. First, on the encoder side, we replace the current "pool-trained" encoder with a "well-trained" one. To achieve this, we train multiple models with different pooler's output dimensions and select the best encoder to replace the current one. Next, on the pooler side, since the pooler and the new encoder have not been trained together, we can fine-tune the pooler on top of the new encoder. This involves training the pooler from its current state while keeping the new encoder frozen, ensuring their compatibility and improving overall performance.

We conduct experiments on seven STS tasks and seven classification tasks. Our proposed training method consistently outperforms all baseline methods across all tasks, for instance, 1.50% to 4.92% improvement over the best baseline method on classification tasks. Remarkably, our method reduces the dimension of sentence embeddings from 768 to 128 with almost no performance loss (from 76.57% to 76.46% on STS tasks). In addition, we validate the effectiveness of the two steps in our proposed method by showing that their average improvement is 1.79% and 1.17% respectively when trained with SimCSE, and 13.16% and 0.83% respectively when trained with Sentence-BERT, on the STS-B dataset.

The key contributions of this paper are:

- We propose customizing the dimension of sentence embeddings by directly modifying the output dimension of the pooler.

- We demonstrate that default dimension of sentence embeddings commonly used in literature is usually suboptimal.

- We discover that the performance loss of low-dimensional sentence embeddings can be divided into the encoder's performance loss and the pooler's performance loss.

- We propose a two-step training method to reduce the two parts of the performance loss separately.

## 2 Sentence Embedding Compressor

Existing sentence representation learning models usually set the dimension of output sentence embeddings as the dimension of hidden states $D$, i.e., $D = 768$ for base models and $D = 1,024$ for large models. However, it is worth noting that the default dimension may not always be optimal. Traditional dimension reduction methods, such as Principle Component Analysis (PCA) (Abdi and Williams, 2010), Isomap (Tenenbaum et al., 2000), and Locally Linear Embedding (LLE) (Roweis and Saul, 2000), are not suitable for our purpose here due to the following reasons: (1) Our objective is to conduct a comprehensive study on the impact of dimension, whereas these methods can only reduce dimension rather than increase it; (2) These methods typically require access to the entire evaluation set before executing the algorithms, which may not be feasible in practical scenarios like online or streaming settings; (3) Utilizing these methods as a post-processing step will introduce extra computational overhead, which, to some extent, contradicts our initial goal of dimension reduction.

We propose a straightforward and efficient approach to modify the dimension of sentence embeddings. As illustrated in the left half of Figure 1, a sentence representation learning model, such as BERT (Devlin et al., 2018) or RoBERTa (Liu et al., 2019), usually includes a pooler on top of the final hidden state of the [CLS] token. This pooler consists of a fully connected layer and a non-linear activation function. Initially, the pooler's purpose is to condense information from the input sentence into a fixed-sized representation without changing

the embedding's dimension. However, we can alter the output dimension of the fully connected layer in the pooler from the default $D$ to a customizable value of $d$. As a result, the pooler now serves as a compressor for sentence embeddings. Unlike conventional dimension reduction techniques, our method can generate sentence embeddings of any dimension. Furthermore, it does not require prior access to the entire evaluation set and has minimal impact on computational overhead.

## 3 The Impact of the Dimension of Sentence Embeddings

We conduct a study to examine the impact of the dimension of sentence embeddings on the performance of various downstream tasks. We select RoBERTa$_{base}$ (Liu et al., 2019) as the sentence representation learning model and made its output dimension configurable. We utilize the unsupervised SimCSE (Gao et al., 2021) as the training method, which takes an input sentence and predicts itself in a contrastive objective with dropout used as noise. Similar to SimCSE, we train the model on one million randomly sampled sentences from English Wikipedia, then apply the model to the following downstream tasks: (1) TREC, a question classification dataset containing 500 labeled questions in the test set with 6 class labels; (2) STS-B, a semantic textual similarity dataset containing 1,379 sentence pairs in the test set with 5 similarity grades; (3) CR, a binary sentiment classification dataset containing 3,773 sentences; (4) MRPC, a binary paraphrase detection dataset containing 1,726 sentence pairs in the test set.

The results of the accuracy / Spearman's correlation of SimCSE-RoBERTa$_{base}$ on the four datasets are presented in Figure 2. The sentence embedding dimension $d$ ranges from 2,048 to 4, with the default value being $D = 768$. As the sentence embedding dimension increases from 768, the performance consistently remains stable across all four datasets. However, when the dimension decreases from 768, we observe distinct patterns in the performance curves: The performance on TREC (the red curve) continuously decreases, and the performance on STS-B and CR (the yellow and the blue curves) initially remains stable, then drops sharply. Conversely, the performance on MRPC (the green curve) remains consistently stable throughout.

It can be concluded that the optimal[1] dimen-

---
[1] Although there is no strict definition for "optimal", it can

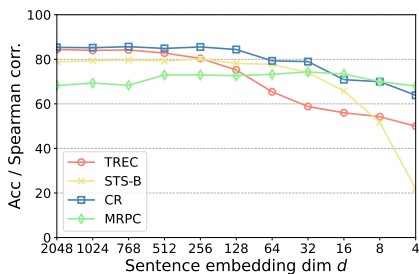

Figure 2: The results of accuracy / Spearman's correlation of SimCSE-RoBERTa$_{base}$ on four different datasets. The sentence embedding dimension $d$ is varied from 2,048 to 4 (the default value is $D = 768$).

sion of sentence embeddings varies across different downstream tasks. Specifically, the optimal dimensions for TREC, STS-B, CR, and MRPC are 768, 256, 256, and 16, respectively. One possible explanation for this variation is that downstream tasks exhibit different levels of difficulty, requiring varying amounts of information to be stored in embeddings to achieve the best performance. This observation motivates further exploring the dimensionality of sentence embeddings, particularly to enhance model performance in low-dimension scenarios.

## 4 The Proposed Approach

### 4.1 Performance Loss Decomposition

According to the result presented in Figure 2, the performance of sentence embeddings on most tasks declines as their dimension decreases. The primary reason for the performance loss is that sentence embeddings become too short to retain sufficient information for downstream tasks. Nevertheless, given that the entire model is trained end-to-end, it is intriguing to examine whether the encoder component is affected when the output dimension of the pooler decreases. Therefore, we denote the final hidden state of [CLS] as the "output of the encoder" and utilize it as the sentence embedding for downstream tasks.

The results of using the encoder's output and the pooler's output as sentence embeddings on the STS-B dataset are presented in Figure 3. Interestingly, when the pooler's output dimension $d$ decreases, the encoder's performance consistently declines for all four models, even though the dimension of the encoder's output remains unchanged. This finding

---
generally be understood as the dimension that maintains the best performance while being as small as possible.

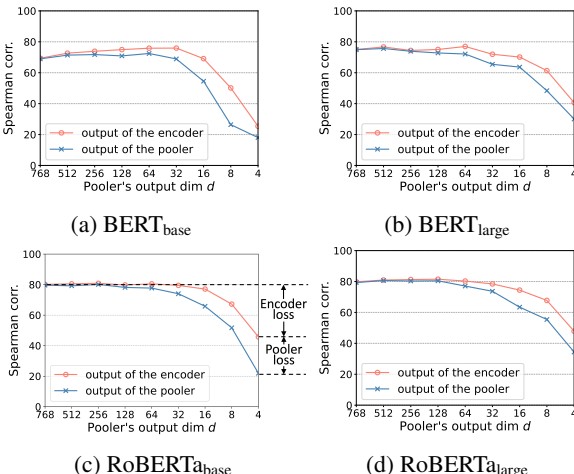

(a) BERT$_{base}$  (b) BERT$_{large}$

(c) RoBERTa$_{base}$  (d) RoBERTa$_{large}$

Figure 3: The results of using the output of the encoder (red curves) and the output of the pooler (blue curves) as sentence embeddings on the STS-B dataset. The training method is SimCSE and the sentence encoders are BERT$_{base}$, BERT$_{large}$, RoBERTa$_{base}$ and RoBERTa$_{large}$, respectively. Figure 3c illustrates that the performance loss can be divided into the performance loss of the encoder and the performance loss of the pooler.

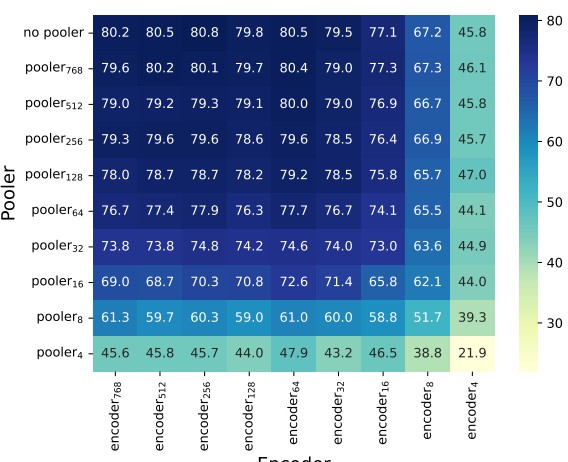

Figure 4: The results of Spearman's correlation of all possible combinations of encoders and poolers on the STS-B dataset. See Section 4.2 for details.

suggests that the performance loss is not solely attributed to the decrease of the pooler's output dimension but also to a deterioration in the quality of the encoder's output.

Figure 3c illustrates that the performance loss can thus be divided into two components: performance loss caused by the encoder and performance loss caused by the pooler. This decomposition of performance loss enables an in-depth understanding of the model's behavior in low-dimensional scenarios. Furthermore, it provides valuable insights into strategies that can improve the model's performance when working with smaller sentence embedding dimensions: By separately addressing the performance loss of the encoder and the pooler, we can effectively enhance the performance of the entire model and subsequently combine the two modules to achieve better outcomes.

## 4.2 Reducing Performance Loss of the Encoder

Figure 3 indicates that the encoder's performance declines noticeably as the pooler's output dimension $d$ decreases. It is worth noting that the encoder's architecture remains unchanged regardless of $d$. As a result, we can easily replace a "pool-trained" encoder with a "well-trained" one to evaluate if the model's overall performance can be enhanced. We conduct end-to-end training of the

SimCSE-RoBERTa$_{base}$ using different pooler output dimensions $d$ (ranging from 768 to 4). This results in a model consisting of encoder$_d$ and pooler$_d$. We then combine each possible encoder$_i$ and pooler$_j$, and utilize the new model encoder$_i$ + pooler$_j$ to generate sentence embeddings.

In Figure 4, each cell in the heatmap represents the Spearman's correlation of a combined model on the STS-B dataset. Replacing the encoder with a superior one can usually substantially enhance the model's overall performance. For instance, the initial performance of the end-to-end training model with $d = 16$ (encoder$_{16}$ + pooler$_{16}$) is 65.8, but it can be further elevated to 72.6 by replacing encoder$_{16}$ with encoder$_{64}$.

We thus propose a method to reduce the performance loss of the encoder, which is illustrated in Figure 5a. Given the target dimension $d$, we first train a sentence representation learning model with the pooler's output dimension being $d$, which consists of encoder$_d$ and pooler$_d$. Meanwhile, we train multiple models with other pooler's output dimensions (e.g., 512, 256, ...). From these models, we select the dimension $opt$ that yields the optimal performance for encoder$_{opt}$ on a validation set. Lastly, we replace the original encoder$_d$ with encoder$_{opt}$ to improve the overall performance.

## 4.3 Reducing Performance Loss of the Pooler

Unlike the encoder, replacing pooler$_d$ with a different pooler$_{d'}$ is not feasible since the output dimension of the pooler must be exactly the target dimension $d$. It is important to note that pooler$_d$ is

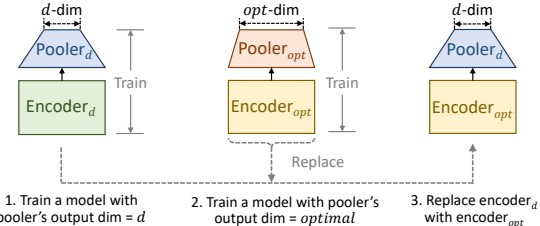

(a) Reducing performance loss of the encoder

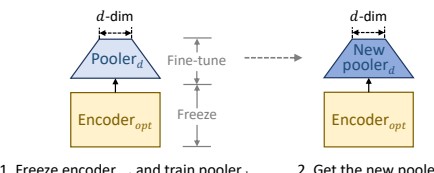

(b) Reducing performance loss of the pooler

Figure 5: Illustration of reducing performance loss of the encoder and the pooler for a sentence representation learning model.

**Algorithm 1:** Two-Step Training Approach

---

**Input:** An unsupervised training corpus $T$, a validation dataset $E$, the target dimension $d$, a base sentence representation learning model $M$ with customizable output dimension;

**Output:** A well-trained sentence representation learning model with output dimension $d$;

// Step 1: Reducing the encoder's loss
1 Determine the set of candidate dimensions $\mathcal{D}$;
2 Train $M$ with $out\_dim = d$ on $T$ and obtain pooler$_d$;
3 **for** $d' \in \mathcal{D}$ **do**
4     Train $M$ with with $out\_dim = d'$ on $T$ and obtain encoder$_{d'}$;
5     Evaluate encoder$_{d'}$ on $E$;
6 Select the best encoder from $\{\text{encoder}_{d'}\}_{d' \in \mathcal{D}}$ and denote it as encoder$_{opt}$;

// Step 2: Reducing the pooler's loss
7 Concatenate encoder$_{opt}$ and pooler$_d$;
8 Fine-tune pooler$_d$ on $T$ with encoder$_{opt}$ frozen, and obtain new-pooler$_d$;
9 **return** encoder$_{opt}$ + new-pooler$_d$

---

trained jointly with encoder$_d$ rather than the current encoder$_{opt}$, which implies that the parameters of pooler$_d$ may not be optimal for encoder$_{opt}$. Therefore, as illustrated in Figure 5b, we freeze the parameters of encoder$_{opt}$ and only fine-tune pooler$_d$, until the model achieves the optimal performance.

Here, we would like to emphasize the following points: (1) The parameters of encoder$_{opt}$ should remain unchanged, as encoder$_{opt}$ is already the optimal encoder. If encoder$_{opt}$ is fine-tuned together with pooler$_d$, we would revert to the initial end-to-end training scenario, which has been shown to yield suboptimal performance. (2) Pooler$_d$ should not be trained from scratch with randomly initialized parameters but rather fine-tuned starting from its current parameters, as it provides an excellent starting point. Our experimental results also validate that fine-tuning from the current parameters outperforms training from randomly initialized parameters.

### 4.4 A Two-Step Training Algorithm

Our proposed two-step training approach is outlined in Algorithm 1. The algorithm consists of two steps. In the first step, the primary objective is to acquire pooler$_d$ (line 2) and the optimal encoder encoder$_{opt}$ (line 6). Subsequently, the second step involves fine-tuning pooler$_d$ while keeping encoder$_{opt}$ frozen (line 8).

The time complexity analysis of Algorithm 1 is as follows. We use $C$ to denote the time required for training the entire model $M$ once. In step 1, we train the model $M$ for a total of $|\mathcal{D}| + 1$ times,

resulting in a complexity of $|\mathcal{D}|C$. The time complexity of the encoder evaluation is negligible compared to the training process. In step 2, the encoder is frozen, and only the pooler undergoes training. Since the pooler is merely a fully connected layer, while the encoder is typically much more complex than the pooler, the fine-tuning time for the pooler is negligible compared to $C$. Therefore, the overall time complexity of Algorithm 1 is $|\mathcal{D}|C$.

Our algorithm generally requires more running time when the candidate dimension set $\mathcal{D}$ is larger. However, a larger pool of $\mathcal{D}$ will also increase the probability that encoder$_{opt}$ performs better, thereby improving the final performance.

## 5 Experiments

### 5.1 Experimental Setup

We evaluate our proposed two-step training algorithm on two types of datasets:

- STS datasets. We include seven STS datasets in our experiments: STS 2012-2016 (Agirre et al., 2012, 2013, 2014, 2015, 2016), STS Benchmark (Cer et al., 2017), and SICK-Relatedness (Marelli et al., 2014). Each dataset consists of sentence pairs and their corresponding ground-truth similarity scores. We use Spearman's correlation to evaluate the predicted results of our method and all baseline methods on the test set.

- Sentence classification datasets. These includes MR (Pang and Lee, 2005), CR (Hu

| Pooler's output dim $d$ | 768 | 512 | 256 | 128 | 64 | 32 | 16 | 8 |
|---|---|---|---|---|---|---|---|---|
| Encoder$_d$ + pooler$_d$ (end-to-end training) | 79.62 | 79.22 | 79.11 | 78.17 | 77.72 | 74.02 | 65.82 | 51.72 |
| Encoder$_{opt}$ + pooler$_d$ (After step 1) | 80.08 +0.46 | 79.26 +0.04 | 79.11 +0.00 | 78.66 +0.49 | 77.94 +0.22 | 74.78 +0.76 | 69.62 +3.80 | 60.27 +8.55 |
| Encoder$_{opt}$ + new-pooler$_d$ (After step 2) | 80.49 +0.41 | 80.25 +0.99 | 79.32 +0.21 | 79.93 +1.27 | 78.03 +0.09 | 75.20 +0.42 | 71.71 +2.09 | 64.15 +3.88 |

Table 1: The results of Spearman's correlation (in %) of our proposed algorithm on the STS-B dataset using the contrastive loss in SimCSE as the training objective. The base model is RoBERTa$_{base}$. (1) The first block is the results of end-to-end training. (2) The second block results from step 1 of our proposed algorithm. The numbers in the second line are the absolute improvement over the first block. (3) The third block results from step 2 of our proposed algorithm. The numbers in the second line are the absolute improvement over the second block. $opt = 256$ for this experiment according to the first row of Figure 4.

| Pooler's output dim $d$ | 768 | 512 | 256 | 128 | 64 | 32 | 16 | 8 |
|---|---|---|---|---|---|---|---|---|
| Encoder$_d$ + pooler$_d$ (end-to-end training) | 70.12 | 69.92 | 63.80 | 60.12 | 56.51 | 52.84 | 49.49 | 39.29 |
| Encoder$_{opt}$ + pooler$_d$ (After step 1) | 73.50 +3.38 | 69.92 +0.00 | 73.34 +9.54 | 73.53 +13.41 | 72.50 +15.99 | 71.46 +18.62 | 68.36 +18.87 | 64.78 +25.49 |
| Encoder$_{opt}$ + new-pooler$_d$ (After step 2) | 73.61 +0.11 | 70.14 +0.22 | 73.95 +0.61 | 73.81 +0.28 | 73.12 +0.62 | 72.88 +1.42 | 70.58 +2.22 | 65.92 +1.14 |
| Encoder$_d$ (pooler$_d$ used only in training) | 73.47 | 73.83 | 66.66 | 63.32 | 61.69 | 57.45 | 56.42 | 47.37 |

Table 2: The results of Spearman's correlation (in %) of our proposed algorithm on the STS-B dataset using the softmax classification loss in Sentence-BERT as the training objective. The first three blocks are similar as in Table 1. The last block is the result of using encoder$_d$ + pooler$_d$ for end-to-end training but only encoder$_d$ for inference. The base model is RoBERTa$_{base}$. $opt = 512$ for this experiment according to the last block.

and Liu, 2004), SUBJ (Pang and Lee, 2004), MPQA (Wiebe et al., 2005), SST (Socher et al., 2013), TREC (Voorhees and Tice, 2000), and MRPC (Dolan and Brockett, 2005). A logistic regression classifier is trained on top of (frozen) sentence embeddings. Each dataset consists of sentences and their class labels. Accuracy is used as the evaluation metric. We follow default configurations from SentEval[2].

We use three traditional dimension reduction methods as baseline methods, including Principle Component Analysis (PCA), Isomap (Tenenbaum et al., 2000), and Locally Linear Embedding (LLE) (Roweis and Saul, 2000). PCA is a linear dimension reduction method, while Isomap and LLE are nonlinear. We use the embeddings of the first 2,000 sentences from the unsupervised English Wikipedia (Gao et al., 2021) as training data for these models. In addition, we also compare our

[2]https://github.com/facebookresearch/SentEval

method to the direct end-to-end training method using SimCSE (Gao et al., 2021).

## 5.2 Result of the Proposed Approach

The results of Spearman's correlation for our proposed method on the STS-B dataset are presented in Tables 1 and 2. We select RoBERTa$_{base}$ as the base model for both experiments. Table 1 presents the result of using the contrastive loss in SimCSE as the training objective (see Section 3 for training details). Table 2 presents the result of using the softmax classification loss in Sentence-BERT as the training objective. Specifically, following Sentence-BERT, we use SNLI and MNLI datasets as the training data. For a pair of premise and hypothesis in SNLI/MNLI denoted as $u$ and $v$, we first calculate their sentence embeddings $\boldsymbol{u}$ and $\boldsymbol{v}$, and then concatenate $\boldsymbol{u}$, $\boldsymbol{v}$, and $\boldsymbol{u} - \boldsymbol{v}$, followed by a 3-way softmax classifier. The pooling function is $cls$. The batch size is 64. Other hyperparameters are the same as reported in the Sentence-BERT paper.

| Methods | STS12 | STS13 | STS14 | STS15 | STS16 | STS-B | SICK-R | Avg. |
|---|---|---|---|---|---|---|---|---|
| $d = 768$ (w/o dimension reduction) | | | | | | | | |
| SimCSE-RoBERTa$_{base}$ | 70.16 | 81.77 | 73.24 | 81.36 | 80.65 | 80.22 | 68.56 | 76.57 |
| $d = 128$ | | | | | | | | |
| PCA | 65.79 | 76.83 | 67.84 | 76.99 | 74.93 | 74.73 | 62.22 | 71.33 |
| Isomap | 51.55 | 56.97 | 45.52 | 53.17 | 56.01 | 49.26 | 51.36 | 51.98 |
| LLE | 38.54 | 45.41 | 34.76 | 42.42 | 45.69 | 40.22 | 42.15 | 41.31 |
| SimCSE-RoBERTa$_{base}$ w/ end-to-end training | 69.10 | 79.69 | 70.80 | 78.70 | 79.66 | 78.17 | 68.32 | 74.92 |
| SimCSE-RoBERTa$_{base}$ w/ two-step training | **70.15** | **80.66** | **72.38** | **81.74** | **80.62** | **79.93** | **69.77** | **76.46** |
| $d = 64$ | | | | | | | | |
| PCA | 65.94 | 75.81 | 66.72 | 75.97 | 73.78 | 73.08 | 60.46 | 70.25 |
| Isomap | 49.69 | 54.85 | 43.25 | 49.90 | 53.99 | 46.82 | 49.32 | 49.69 |
| LLE | 33.54 | 42.57 | 32.38 | 38.78 | 40.24 | 36.40 | 37.04 | 37.28 |
| SimCSE-RoBERTa$_{base}$ w/ end-to-end training | 66.29 | 78.76 | 70.55 | **80.18** | 78.33 | 77.72 | 67.85 | 74.24 |
| SimCSE-RoBERTa$_{base}$ w/ two-step training | **68.73** | **80.34** | **71.63** | 79.90 | **79.61** | **78.03** | **68.62** | **75.27** |
| $d = 32$ | | | | | | | | |
| PCA | 65.04 | 72.92 | 64.14 | 73.16 | 71.31 | 69.15 | 58.08 | 67.69 |
| Isomap | 46.36 | 50.89 | 40.23 | 45.92 | 51.00 | 44.61 | 47.00 | 46.57 |
| LLE | 32.33 | 35.37 | 24.99 | 30.81 | 35.84 | 32.37 | 33.09 | 32.11 |
| SimCSE-RoBERTa$_{base}$ w/ end-to-end training | 63.33 | 77.71 | 66.67 | 76.06 | 76.23 | 74.02 | 67.22 | 71.61 |
| SimCSE-RoBERTa$_{base}$ w/ two-step training | **67.89** | **77.80** | **69.77** | **77.66** | **77.38** | **75.20** | **68.26** | **73.42** |
| $d = 16$ | | | | | | | | |
| PCA | 62.75 | 67.67 | 59.97 | 68.13 | 67.23 | 63.46 | 55.53 | 63.53 |
| Isomap | 42.44 | 44.79 | 34.57 | 42.42 | 45.63 | 40.33 | 43.43 | 41.94 |
| LLE | 28.55 | 33.95 | 23.66 | 29.67 | 34.13 | 30.82 | 32.01 | 30.40 |
| SimCSE-RoBERTa$_{base}$ w/ end-to-end training | 54.16 | 67.31 | 55.85 | 63.92 | 70.70 | 65.82 | 63.64 | 63.06 |
| SimCSE-RoBERTa$_{base}$ w/ two-step training | **64.82** | **75.16** | **65.32** | **74.87** | **74.25** | **71.71** | **66.15** | **70.33** |
| $d = 8$ | | | | | | | | |
| PCA | 53.31 | 56.66 | 51.23 | 59.86 | 60.52 | 52.58 | 49.85 | 54.86 |
| Isomap | 38.54 | 39.11 | 30.79 | 39.32 | 41.48 | 35.94 | 37.98 | 37.59 |
| LLE | 30.01 | 33.85 | 22.86 | 33.06 | 37.88 | 28.66 | 33.50 | 31.40 |
| SimCSE-RoBERTa$_{base}$ w/ end-to-end training | 50.92 | 51.26 | 43.27 | 59.03 | 58.84 | 51.72 | 58.03 | 53.30 |
| SimCSE-RoBERTa$_{base}$ w/ two-step training | **60.89** | **65.25** | **59.01** | **65.86** | **66.84** | **64.15** | **59.61** | **63.09** |

Table 3: The results of Spearman's correlation (in %) on seven STS datasets. Each block corresponds to a specific dimension of sentence embeddings. The highest numbers across all methods are highlighted.

In Tables 1 and 2, the first block shows the results of encoder$_d$ + pooler$_d$, representing the end-to-end training approach. In the second block, we present the results of encoder$_{opt}$ + pooler$_d$, corresponding to step 1 of our proposed algorithm. The third block results from encoder$_{opt}$ + new-pooler$_d$, corresponding to step 2 of our proposed algorithm. In addition, in Table 2, we also present the result of Encoder$_d$ as the last block, in which encoder$_d$ + pooler$_d$ is used in end-to-end training but only encoder$_d$ is used in inference. Note that $opt = 256$ in Table 1 while $opt = 512$ in Table 2. The absolute improvement achieved by step 1 and step 2 is also presented.

We observe that step 1 and step 2 of our method both yield significant enhancement to the model's performance. The average absolute improvement achieved by step 1 and step 2 is 1.79% and 1.17% respectively using SimCSE, and is 13.16% and 0.83% respectively using Sentence-BERT. Notably, the improvement brought about by step 1 surpasses

that of step 2 for two primary reasons. First, step 2 faces a greater challenge in improving the model since step 1 has already substantially enhanced its performance. Second, the encoder is typically more complicated than the pooler, which offers greater potential for step 1 to improve the performance. Moreover, the improvement is particularly pronounced when the dimension $d$ is smaller, as the model has more room for improvement in low-dimensional scenarios.

## 5.3 Comparison with Baseline Methods

The results of comparing with baseline methods on STS tasks and classification tasks are presented in Tables 3 and 4, respectively. Each block corresponds to a specific dimension of sentence embeddings. We do not show the results of $d = 512$ and $d = 256$ because their results are quite close to $d = 768$. It is evident that our method consistently achieves the best performance across almost all

| Methods | MR | CR | SUBJ | MPQA | SST | TREC | MRPC | Avg. |
|---|---|---|---|---|---|---|---|---|
| $d = 768$ (w/o dimension reduction) | | | | | | | | |
| SimCSE-RoBERTa$_{base}$ | 81.04 | 87.74 | 93.28 | 86.94 | 86.60 | 84.60 | 73.68 | 84.84 |
| $d = 128$ | | | | | | | | |
| PCA | 72.76 | 79.93 | 84.33 | 79.03 | 76.93 | 68.00 | 62.13 | 74.73 |
| Isomap | 58.82 | 65.46 | 72.20 | 68.03 | 57.39 | 31.20 | 67.13 | 60.03 |
| LLE | 58.07 | 65.41 | 69.64 | 68.99 | 57.66 | 28.00 | 66.49 | 59.18 |
| SimCSE-RoBERTa$_{base}$ w/ end-to-end training | 77.52 | **84.40** | 89.99 | 81.86 | 82.26 | 75.20 | 72.64 | 80.55 |
| SimCSE-RoBERTa$_{base}$ w/ two-step training | **79.00** | 84.11 | **91.20** | **84.35** | **83.69** | **80.00** | **73.91** | **82.32** |
| $d = 64$ | | | | | | | | |
| PCA | 70.69 | 78.29 | 80.46 | 76.28 | 73.69 | 56.80 | 61.20 | 71.06 |
| Isomap | 58.06 | 64.85 | 70.41 | 67.61 | 58.98 | 32.60 | 67.36 | 59.98 |
| LLE | 55.90 | 63.81 | 64.75 | 68.98 | 55.52 | 24.80 | 66.49 | 57.18 |
| SimCSE-RoBERTa$_{base}$ w/ end-to-end training | 72.82 | 79.31 | 84.65 | 77.82 | 76.55 | 65.40 | 73.28 | 75.69 |
| SimCSE-RoBERTa$_{base}$ w/ two-step training | **76.52** | **83.66** | **90.24** | **80.90** | **81.93** | **75.60** | **75.42** | **80.61** |
| $d = 32$ | | | | | | | | |
| PCA | 66.02 | 74.02 | 76.44 | 71.63 | 71.22 | 48.20 | 64.28 | 67.40 |
| Isomap | 57.96 | 66.67 | 71.15 | 68.68 | 58.05 | 34.00 | 67.07 | 60.51 |
| LLE | 52.98 | 63.79 | 64.01 | 68.77 | 53.21 | 22.60 | 66.49 | 55.98 |
| SimCSE-RoBERTa$_{base}$ w/ end-to-end training | 72.15 | 78.67 | 79.53 | **77.10** | 74.42 | 58.80 | 72.32 | 73.28 |
| SimCSE-RoBERTa$_{base}$ w/ two-step training | **73.68** | **78.97** | **86.49** | 76.30 | **75.56** | **69.60** | **73.91** | **76.36** |
| $d = 16$ | | | | | | | | |
| PCA | 62.21 | 72.19 | 74.84 | 70.71 | 66.06 | 40.80 | 65.72 | 64.65 |
| Isomap | 53.69 | 65.59 | 68.60 | 68.83 | 58.81 | 34.20 | 66.03 | 59.39 |
| LLE | 52.86 | 63.76 | 62.18 | 68.77 | 51.95 | 20.20 | 66.61 | 55.19 |
| SimCSE-RoBERTa$_{base}$ w/ end-to-end training | 65.07 | 70.86 | 80.65 | **74.19** | 68.31 | 54.00 | 70.39 | 69.07 |
| SimCSE-RoBERTa$_{base}$ w/ two-step training | **69.21** | **74.68** | **84.24** | 74.09 | **73.37** | **59.20** | **73.33** | **72.59** |
| $d = 8$ | | | | | | | | |
| PCA | 58.85 | 67.82 | 67.98 | 66.61 | 63.70 | 36.60 | 64.16 | 60.82 |
| Isomap | 55.17 | 65.40 | 67.16 | 68.96 | 61.29 | 27.00 | 66.84 | 58.83 |
| LLE | 50.16 | 63.76 | 60.98 | 68.77 | 50.19 | 18.80 | 66.49 | 54.16 |
| SimCSE-RoBERTa$_{base}$ w/ end-to-end training | 66.50 | 69.96 | 77.62 | 72.26 | **70.18** | 44.20 | 69.97 | 67.24 |
| SimCSE-RoBERTa$_{base}$ w/ two-step training | **66.55** | **71.07** | **78.74** | **73.38** | 68.59 | **51.00** | **71.83** | **68.74** |

Table 4: The results of accuracy (in %) on seven sentence classification datasets. Each block corresponds to a specific dimension of sentence embeddings. The highest numbers across all methods are highlighted.

cases. For example, when $d = 32$, our method outperforms the best traditional dimension reduction method by 5.73% and 8.96% on average for STS tasks and classification tasks, respectively. Similar to Table 1, the improvement becomes more significant when the dimension decreases.

It is exciting to observe that our method exhibits minimal performance degradation when $d$ decreases from 768 to 128 (from 76.57% to 76.46% on STS tasks), indicating that sentence embeddings can be effectively compressed to just $1/6$ of the original size with almost no loss in performance. We also observe that, despite being a linear dimension duction method, PCA consistently outperforms the other two nonlinear dimension reduction methods.

## 6 Related Work

**Sentence Representation Learning**
Researchers have proposed numerous methods

for sentence representation learning. For example, SBERT (Reimers and Gurevych, 2019) uses siamese and triplet network structures to derive semantically meaningful sentence embeddings that can be compared using cosine-similarity. DPR (Karpukhin et al., 2020) uses embeddings for information retrieval, which are learned from a small number of questions and passages by a simple dual-encoder framework. SimCSE (Gao et al., 2021) takes an input sentence and predicts itself in a contrastive objective with dropout used as noise. Building upon SimCSE, DiffCSE (Chuang et al., 2022) and ESimCSE (Wu et al., 2021) further enhance the method by improving the sampling approach. InstructOR (Su et al., 2022) embeds every text together with instructions explaining the use case, which can generate text embeddings for different downstream tasks and domains without further training. However, these works overlook the study of how the dimension of sentence embeddings im-

pacts the model's performance. In contrast, our work focuses on enhancing the performance of sentence embeddings in low-dimensional scenarios. Our proposed training algorithm can be employed in conjunction with any Transformers-based language models and the aforementioned sentence representation learning methods.

**Dimension Reduction**

Dimension reduction is a technique that reduces the number of features in a dataset while preserving the essential information. For instance, PCA (Abdi and Williams, 2010) is a linear dimensionality reduction technique that finds a new set of uncorrelated variables (principal components) by projecting the data onto a lower-dimensional subspace while maximizing the variance. Isomap (Tenenbaum et al., 2000) is a nonlinear dimensionality reduction algorithm that preserves the geodesic distances between data points, creating a low-dimensional embedding that captures the intrinsic structure of the data manifold. LLE (Roweis and Saul, 2000) is a nonlinear dimensionality reduction method that seeks to preserve local relationships between neighboring data points, constructing a lower-dimensional representation based on linear combinations of these neighbors. However, as discussed earlier, these traditional dimension reduction methods are not suitable for our task as they require access to the entire evaluation set in advance and they introduce additional computation cost. Another related work is (Yin and Shen, 2018), which theoretically studies the optimal dimension of word embeddings.

## 7   Conclusion

This paper presents a comprehensive and empirical study on the dimensionality of sentence embeddings. First, we propose customizing the dimension of sentence embeddings by directly modifying the pooler's output dimension. Subsequently, we demonstrate that the default dimension (768 or 1,024) of sentence embeddings commonly used in the literature are usually suboptimal. To enhance the performance of low-dimensional sentence embeddings, we decompose the performance loss into the encoder's loss and the pooler's loss. We then introduce a two-step training method that separately addresses the two parts of the performance loss. Experimental results demonstrate that our proposed training method consistently enhances the performance of sentence embeddings with low dimensions across all tasks.

**Limitations**

In this paper, we aim to thoroughly comprehend the dimensionality of sentence embeddings, focusing primarily on empirical and experimental aspects. However, note that there remain unanswered questions concerning the dimension of sentence embeddings, especially from a theoretical perspective, which we leave as future work.

Firstly, Figure 3 illustrates that reducing the output dimension of the pooler leads to worse performance of the encoder. One possible explanation is that when the dimension is too small, sentence embeddings are unable to capture all the information in sentences, resulting in an inadequate representation of sentences. Consequently, the quality of the back-propagated signal from the pooler diminishes, which hinders the effective training of the encoder. However, a theoretical understanding of this phenomenon is currently lacking.

Secondly, as depicted in Figure 4, replacing the current encoder $\text{encoder}_d$ with a "well-trained" $\text{encoder}_{opt}$ improves the performance of $\text{pooler}_d$'s output. It should be noted that $\text{encoder}_{opt}$ and $\text{pooler}_d$ are not trained jointly, which implies that the output embedding space of $\text{encoder}_{opt}$ and the input embedding space of $\text{pooler}_d$ are not aligned. This suggests that a simple concatenation of $\text{encoder}_{opt}$ and $\text{pooler}_d$ might not produce embeddings with physical meaning. However, experimental results demonstrate the effectiveness of this substitution strategy. The exact reason behind the improvement remains unknown.

Lastly, an intriguing relationship exists between PCA and the pooler of language models. While PCA applies a linear transformation to sentence embeddings, the pooler applies a linear transformation followed by a nonlinear function (`tanh` in our model). Notably, we also experiment with removing the nonlinear function from the pooler, and find that the model's performance did not significantly change. Therefore, the pooler can be considered as a rough approximation of a PCA layer, and we indeed discover that PCA is the most effective dimension reduction approach among the baseline methods. Given that the linear transformation in PCA aims to project data onto a low-dimensional space while maximizing the variance, it is intriguing to investigate how the pooler projects sentence embeddings and whether a theoretical connection exists between the linear transformation in PCA and the pooler.

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
