# OpenReview forum: "On the Dimensionality of Sentence Embeddings"
_EMNLP/2023/Conference — EMNLP 2023 Findings_

### Official Review · Reviewer_5JDc · 2023-08-04

**Typos Grammar Style And Presentation Improvements:** Line 250 - Figure 3c should be Figure 3.
**Soundness:** 3

**Excitement:**

2: Mediocre: This paper makes marginal contributions (vs non-contemporaneous work), so I would rather not see it in the conference.

**Missing References:**

NA

**Paper Topic And Main Contributions:**

Study the dimensionality of sentence embedding and propose a way to get lower dimensional sentence embedding without much loss of performance on downstream tasks.

**Questions For The Authors:**

1. The SimCSE paper found the last second layer performs better in STS tasks when doing unsupervised training. If so, the pooling will not affect the sentence embedding dimensions. Do you have similar finding and how it affects your experiments?

**Reasons To Accept:**

1. The paper is easy to follow.

2. Reasonable performance after sentence embedding compression (by dimensions).

**Reasons To Reject:**

1. The idea is not actually less interesting. It is widely known that adjusting the dimensions be more efficient and the performance may not downgrade much when the compression is not too aggressive.

2. The proposed method is closer to training tricks. It shows some phenomena but there is no solid explanation of why it happens or presentation of the author's insights towards it.

3. The author shows multiple variants of embedding dimensions and optimal dimensions. It would be more meaningful if I can know how to choose the optimal dimension besides trying all combinations.

**Reproducibility:**

3: Could reproduce the results with some difficulty. The settings of parameters are underspecified or subjectively determined; the training/evaluation data are not widely available.

**Reviewer Confidence:**

5: Positive that my evaluation is correct. I read the paper very carefully and I am very familiar with related work.

---

> ### Author Rebuttal · Authors · 2023-08-28
>
> Thanks so much for your valuable review. Our response to your comments is as follows.
>
> ----
>
> **Q1**: *The idea is not interesting. It is widely known that adjusting the dimensions is more efficient and the performance may not downgrade much when the compression is not too aggressive.*
>
> **A1**: Our contribution in this paper is three-fold: (1) the default dimension of sentence embedding is not optimal and can thus be reduced (this is what you mentioned); (2) the performance loss of low-dimensional sentence embeddings can be divided into two parts; (3) we propose a two-step training method to reduce the two parts of the performance loss separately. What you mentioned is only the Section 3 of this paper, while most of our contribution is presented in Sections 4 and 5.
>
> ----
>
> **Q2**: *The proposed method is closer to training tricks. It shows some phenomena but there is no solid explanation of why it happens or presentation of the author's insights towards it.*
>
> **A2**: First, training tricks are actually important. RoBERTa is also some training tricks for BERT but it is undoubtedly a highly influential language model.
> Second, as mentioned in the section of limitation, it is true that we have no theoretical explanation for our findings, but we are the first to discover the phenomenon of performance loss decomposition, and we have accordingly proposed a realistic training method. We are not expected to solve all the (theoretical) problems regarding this phenomenon, and we are not supposed to be punished for this.
> Third, we actually have the insight towards the training trick. The reason of proposing the two-step training method is exactly because we found that the performance loss can be decomposed into the encoder’s loss and the pooler’s loss. This finding is critical for improving the performance of language models in the scenario of low-dimension sentence embedding.
> Fourth, we actually have provided our theoretical guess in the section of limitation, which is the connection between the pooler layer and PCA. Further investigation on this problem is beyond the scope of this work and this conference (Empirical Methods in NLP).
>
> ----
>
> **Q3**: *The author shows multiple variants of embedding dimensions and optimal dimensions. It would be more meaningful if I can know how to choose the optimal dimension besides trying all combinations.*
>
> **A3**: First, you can speed up the procedure of searching the optimum in Figure 4 by skipping columns beyond “encoder-32” and rows beyond “pooler-32”, since the performance has significantly dropped when the dimension reduces to 32.
> Second, searching the optimum in Figure 4 is actually not that time-consuming, because you only need to concatenate the encoder and the pooler and then evaluate this combined model. This does not involve training the model (so it is different from the traditional grid search on hyperparameters) and is thus time-efficient.
>
> ----
>
> **Q4**: *The SimCSE paper found the last second layer performs better in STS tasks when doing unsupervised training. If so, the pooling will not affect the sentence embedding dimensions. Do you have similar finding and how it affects your experiments?*
>
> **A4**: We do have similar findings. You can observe from Figure 4 that, the row of “no pooler” is better than the row of “pooler-768” in most cases. But this finding does not affect our experiment, because in our model architecture, the pooler layer is used to adjust the embedding dimension from 768 to the target dimension d, so the pooler layer is mandatory in our model. This is different from SimCSE where the shape of pooler layer is 768*768, so the pooler is optional for SimCSE depending on whether it can improve the model performance.
>
> ----
>
> **Q5**: *Typo: Line 250 - Figure 3c should be Figure 3.*
>
> **A5**: This is not a typo. Please note that in Figure 3c we explicitly annotate  the encoder loss and the pooler loss.

---

### Official Review · Reviewer_s3jK · 2023-08-04

**Soundness:** 4

**Excitement:**

3: Ambivalent: It has merits (e.g., it reports state-of-the-art results, the idea is nice), but there are key weaknesses (e.g., it describes incremental work), and it can significantly benefit from another round of revision. However, I won't object to accepting it if my co-reviewers champion it.

**Paper Topic And Main Contributions:**

This article presents a study of the impact of dimension of word and sentence embeddings in popular language, including transformers.

**Questions For The Authors:**

Do you think the improvement of the results with lower embeddings is due to the fact that the training of the encoder and pooler at the same time (in high dimension) performs bad in general?
Or do you think, it is because lower dimensional embeddings allow to ``condense the information''  (or maybe both?)
Would there be an experimental way of trying to answer this question?

**Reasons To Accept:**

- The finding are interesting (lower dimensions are better for the tasks considered)

- The presentation of the article (motivation, experiments, discussion) is clear

- The two step training method (separate the encoder part from the pooler part) is simple to implement, and easy to reproduce.

**Reasons To Reject:**

- The mention and references to similar existing related work (on the impact of dimension of the embeddings for popular language models, and not only dimension reduction in general) is almost absent. If there is few or none, then it should be made more clearly by the authors.

- Although I find the main message of the article interesting, the types of experiments are limited (the ones conducted are thoroughly conducted, but there is no other task than STS and text classification), this limits the impact of the study. It would have been very interesting to see the results for various types of tasks, such as named entity recognition, or language translation. I do not expect the authors to perform the experiments for every NLP task, but at least one or two other task would help greatly to convince the reader.


**Reproducibility:**

3: Could reproduce the results with some difficulty. The settings of parameters are underspecified or subjectively determined; the training/evaluation data are not widely available.

**Reviewer Confidence:**

4: Quite sure. I tried to check the important points carefully. It's unlikely, though conceivable, that I missed something that should affect my ratings.

**Typos Grammar Style And Presentation Improvements:**

Title of section 2: Comperssor -> Compressor

---

> ### Author Rebuttal · Authors · 2023-08-28
>
> Thanks so much for your valuable review. Our response to your comments is as follows.
>
> ----
>
> **Q1**: *The mention and references to similar existing related work is almost absent.*
>
> **A1**: To our best knowledge, we are not aware of such work specifically discussing the impact of output embedding dimension of BERT-like language models. We are still actively surveying the literature and will add the related work in the future.
>
> ----
>
> **Q2**: *It would be interesting to see the results for various types of tasks, such as named entity recognition, or language translation.*
>
> **A2**:  We followed your comment and conducted experiments on the IR task (the NER and MT task you mentioned usually do not involve sentence embeddings). The IR task aims to retrieve relevant documents for a given query. This task is a perfect downstream application of sentence embeddings, because the query and documents can be encoded into embeddings, and thus the documents can be sorted according to their similarity with the query embedding. Specifically, we randomly select 200 questions from the NQ dataset (https://ai.google.com/research/NaturalQuestions). For each question, we retrieve top 25 paragraphs from Wikipedia dump (https://github.com/google-research-datasets/Attributed-QA)  that has the highest bm25 score with the question. So the entire document candidate pool has 5k paragraphs. Note that each question in NQ dataset has a ground-truth answer. Therefore, for a given question, any paragraph that contains the ground-truth answer to this question is treated as relevant. The result of top-1 accuracy on NQ dataset for d=128 is as follows:
>
> | Dataset | NQ |
> | --- | --- |
> | RoBERTa-base (d=768, w/o dim reduction) | 40.4 |
> | PCA (d=128) | 36.2 |
> | Isomap (d=128) | 29.1 |
> | LLE (d=128) | 25.4 |
> | RoBERTa-base w/ end-to-end training (d=128) | 38.5 |
> | RoBERTa-base w/ two-step training (d=128) | __40.1__ |
>
> The experiment shows similar result with Tables 2 and 3, which demonstrates that our training method works well for the IR task.
>
> ----
>
> **Q3**: *Do you think the improvement of the results with lower embeddings is due to the fact that the training of the encoder and pooler at the same time (in high dimension) performs bad in general? Or do you think, it is because lower dimensional embeddings allow to ``condense the information'' (or maybe both?)*
>
> **A3**: The reason that training the encoder and pooler at the same time in high dimension performs bad lies in that the dimension is too high and sentence embeddings are overfitting the training data. A reasonably low dimension embedding can restrict the capacity of the model and alleviate the overfitting issue.

---

### Official Review · Reviewer_85p1 · 2023-08-08

**Soundness:** 5

**Excitement:**

4: Strong: This paper deepens the understanding of some phenomenon or lowers the barriers to an existing research direction.

**Paper Topic And Main Contributions:**

This paper studies the dimension size used for sentence embeddings in typical NLP applications. The main premise of this research is that by adjusting the dimensionality of sentence embeddings generated from BERT-like encoder models at different stages of the encoding, one could potentially (a) obtain better semantic representations (sentence embeddings) from the model and (b) reduce the model size with negligible tradeoff in quality.

In this work, the authors use several benchmark tasks to conduct experiments on BERT models, varying the sentence dimension in the pooling layer, and in the layers before the pooling layer in the model. Through the set of experiments, the paper makes several interesting observations about dimension size and impact on task performance as well as on model size. Several learnings and conclusions are presented that would be interesting to the community.

**Questions For The Authors:**

The paper mentions that the default choice of dimension size is suboptimal. It would be beneficial for the reader to understand the history behind this default dimension size. What is the reasoning (from prior work) for choosing those default values?

The experiments in this paper uses SimCSE with the contrastive sentence prediction objective for all observations and conclusions. Are there experiments with other objectives, that lead to different conclusions?

Are there any differences in the hyperparameters tuned across the various experiments?

**Reasons To Accept:**

The paper raises some interesting questions about architectural choices in BERT models, and presents some valuable observations through well-crafted experiments.

Based on the observations about dimensionality before and after the pooling stages, the paper is able to draw conclusions that inform architectural choices for researchers working with these models. By running these experiments across a range of tasks, the paper is able to demonstrate the general applicability of the main messages.



**Reasons To Reject:**

The conclusions of this work are limited to BERT, with a specific pre-training process and objective. As such, it would be interesting to learn
whether the learnings and conclusions hold across models trained with other pretraining objectives and other variations in pre-training methodology.

**Reproducibility:**

4: Could mostly reproduce the results, but there may be some variation because of sample variance or minor variations in their interpretation of the protocol or method.

**Reviewer Confidence:**

4: Quite sure. I tried to check the important points carefully. It's unlikely, though conceivable, that I missed something that should affect my ratings.

**Typos Grammar Style And Presentation Improvements:**

Section 2, title: "Sentence Embedding Comperssor" -> "Sentence Embedding Compressor"

---

> ### Author Rebuttal · Authors · 2023-08-28
>
> Thanks so much for your valuable review. Our response to your comments is as follows.
>
> ----
>
> **Q1**: *What is the reason (from prior work) for choosing those default values of dimension size?*
>
> **A1**: The reason of choosing 768 or 1024 as the default dimension size for sentence embeddings is simply because the hidden dimension and the output dimension of BERT-base / BERT-large is 768 / 1024. Previous work usually uses BERT-like models as their base model, so they follow this default setting. We will add the explanation in future versions.
>
> ----
>
> **Q2**: *The experiments use SimCSE with the contrastive sentence prediction objective for all observations and conclusions. Are there experiments with other objectives, that lead to different conclusions?*
>
> **A2**: We followed your comment and conducted more experiments using training objectives of Sentence-BERT. Specifically, following S-BERT, we use RoBERTa-base as the base model and SNLI/MNLI datasets as the training data. For a pair of premise and hypothesis in SNLI/MNLI denoted as u and v, we first calculate their sentence embeddings $u$ and $v$, and then concatenate $u$, $v$, and $u-v$, followed by a 3-way softmax classifier. The pooling function is $cls$. The batch size is 64. Other hyperparameters are the same as reported in the S-BERT paper. The result of the softmax training objective is as follows:
>
> | Pooler’s output dimension $d$ | 768 | 512 | 256 | 128 | 64 | 32 | 16 | 8 |
> | ---- | ---- | ---- | ---- | ---- | ---- | ---- | ---- | ---- |
> | Encoder$_d$ + pooler$_d$ (end-to-end training) | 70.12 | 69.92 | 63.80 | 60.12 | 56.51 | 52.84 | 49.49 | 39.29 |
> | Encoder$_d$ only | 73.47 | 73.83 | 66.66 | 63.32 | 61.69 | 57.45 | 56.42 | 47.37 |
> | Encoder$_{opt}$ + pooler$_d$ (after step 1) | 73.50 | 69.92 | 73.34 | 73.53 | 72.50 | 71.46 | 68.36 | 64.78 |
> | Encoder$_{opt}$ + new-pooler$_d$ (after step 2) | 73.61 | 70.14 | 73.95 | 73.81 | 73.12 | 72.88 | 70.58 | 65.92 |
>
> The results above are similar to Table 1. According to the second row (Encoder$_d$ only), $opt$ is set to 512 for this experiment. The results show that our two-step training method is effective for S-BERT with softmax training objective as well.
>
> ----
>
> **Q3**: *Are there any differences in the hyperparameters tuned across the various experiments?*
>
> **A3**: Our proposed method in Algorithm 1 does not have hyperparameters. For the base model (BERT or RoBERTa) and the training objectives (SimCSE or S-BERT), the hyperparameters are the same as the default values in their public implementation codes.

---

### Official Review · Reviewer_wHEz · 2023-08-11

**Soundness:** 3

**Excitement:**

3: Ambivalent: It has merits (e.g., it reports state-of-the-art results, the idea is nice), but there are key weaknesses (e.g., it describes incremental work), and it can significantly benefit from another round of revision. However, I won't object to accepting it if my co-reviewers champion it.

**Paper Topic And Main Contributions:**

The authors proposed a method to tailor the dimensionality of sentence embeddings by directly altering the output dimension of the pooler. They posited that commonly accepted default dimensions for sentence embeddings, as referenced in much of the literature, are not always optimal. In efforts to reduce the dimensionality of sentence embeddings without significant performance drops, they discerned two primary contributors to performance deterioration: losses from the encoder and the pooler. Their findings indicate that the diminished performance of low-dimensional sentence embeddings can be bifurcated into these two distinct losses. As a resolution, they introduce a two-step training strategy for sentence representation models, wherein the encoder and pooler are optimized independently, ensuring minimal performance degradation in reduced-dimension contexts.

**Reasons To Accept:**

- The study offers a thorough empirical examination of sentence embedding dimensionality.
- The method and experiments are clearly presented.
- The empirical outcomes, especially those illustrating the efficacy of the two-step training model for sentence representations, are notably robust. These results underscore the method's ability to enhance the efficacy of reduced-dimension sentence embeddings across varied tasks.
- The paper is well-written and easy to understand.
- The experiments exhibit a wide scope, spanning multiple tasks and datasets.
- The ablation studies provide clarity on the significance of various hyperparameters.

**Reasons To Reject:**

- The evaluation experiments lack of strong baseline comparisons.

**Reproducibility:**

4: Could mostly reproduce the results, but there may be some variation because of sample variance or minor variations in their interpretation of the protocol or method.

**Reviewer Confidence:**

3: Pretty sure, but there's a chance I missed something. Although I have a good feel for this area in general, I did not carefully check the paper's details, e.g., the math, experimental design, or novelty.

---

> ### Author Rebuttal · Authors · 2023-08-27
>
> Thanks so much for your valuable review. Our response to your comments is as follows.
>
> ----
>
> **Q1**: *The evaluation experiments lack of strong baseline comparisons.*
>
> **A1**: We have conducted a comprehensive survey on the literature, but we are not aware of any method specifically working on reducing the dimension of sentence embeddings. As far as we know, this is the first paper focusing on sentence embedding dimension. Therefore, we can only include some traditional dimension reduction methods as the baseline methods. But note that PCA is actually a strong method according to the results in Table 2.

---

### Meta-Review · Area_Chair_xcwF · 2023-09-19

**Recommendation:** 3

**Metareview:**

This paper systematically presents an interesting observation: when training with different dimensionality of a pooler, performance of using directly with the embedding out of transformer (without pooler) will change correspondingly. Thus performance loss could be decomposed into two part, and author proposed a way to select a low dimensionality with better performance. It's an interesting observation but the significance is not prominent.

Pros:

1. Interesting observation.

2. Present it well and easy to follow

Cons:

1. After running the proposed 2-stage "multiple-train + plugin replacement" algorithm, the only benefit is for pooler to be from D to d, which is less than 1% or even less computation in the whole model. And the performance is not improved a lot either. This leads to the question why we want to study the problem.

2. Results are not consistent on all models. Authors agreed that for RoBERTa-base, the effect is not very prominent, but on other models it matters more.

3. The paper only focused on the "loss decomposition" part, but didn't explain why the performance could actually be "improved". For example, the Figure 3a, using a d=32 pooler to co-train and use encoder output give about 5% accuracy improvement, which is much larger than most improvement in the experiments. This effect is not clearly explained.

---

### Decision · Program_Chairs · 2023-10-07

**Decision:**

Accept-Findings

**Comment:**

This paper systematically presents an interesting observation: when training with different dimensionality of a pooler, performance of using directly with the embedding out of transformer (without pooler) will change correspondingly. Thus performance loss could be decomposed into two part, and author proposed a way to select a low dimensionality with better performance. It's an interesting observation but the significance is not prominent.

Pros:

1. Interesting observation.

2. Present it well and easy to follow

Cons:

1. After running the proposed 2-stage "multiple-train + plugin replacement" algorithm, the only benefit is for pooler to be from D to d, which is less than 1% or even less computation in the whole model. And the performance is not improved a lot either. This leads to the question why we want to study the problem.

2. Results are not consistent on all models. Authors agreed that for RoBERTa-base, the effect is not very prominent, but on other models it matters more.

3. The paper only focused on the "loss decomposition" part, but didn't explain why the performance could actually be "improved". For example, the Figure 3a, using a d=32 pooler to co-train and use encoder output give about 5% accuracy improvement, which is much larger than most improvement in the experiments. This effect is not clearly explained.